# A Machine Learning Study of High Robustness Quantum Walk Search Algorithm with Qudit Householder Coins

Hristo Tonchev [1,2,*,†] and Petar Danev [2,†]

1  Institute of Solid State Physics, Bulgarian Academy of Sciences, 72 Tzarigradsko Chaussée, 1784 Sofia, Bulgaria
2  Institute for Nuclear Research and Nuclear Energy, Bulgarian Academy of Sciences, 72 Tzarigradsko Chaussée, 1784 Sofia, Bulgaria
*  Correspondence: htonchev@inrne.bas.bg
†  These authors contributed equally to this work.

**Abstract:** In this work, the quantum random walk search algorithm with a walk coin constructed by generalized Householder reflection and phase multiplier has been studied. The coin register is one qudit with an arbitrary dimension. Monte Carlo simulations, in combination with supervised machine learning, are used to find walk coins that make the quantum algorithm more robust to deviations in the coin's parameters. This is achieved by introducing functional dependence between these parameters. The functions that give the best performance of the algorithm are studied in detail by numerical statistical methods. A thorough comparison between our modification and an algorithm, with coins made using only Householder reflection, shows significant advantages of the former. By applying a deep neural network, we make a prediction for the parameters of an optimal coin with an arbitrary size and estimate the algorithm's stability for such a coin.

**Keywords:** quantum algorithms; quantum random walk; quantum search; qudits; generalized householder reflection; supervised machine learning; neural networks; Monte Carlo simulations

## 1. Introduction

Many examples of the differences between the classical and quantum worlds can be given. The Quantum Random Walk (QRW) [1] is one of them. Quantum interferences allow quadratically faster graph traversal compared to the classical walk [2]. This was initially tested on simple structures such as line [3] and circle [3]. Later, QRW was used to study more complex structures, such as square and hexagonal grids [4,5], cylinders [6], torus [4], and hypercubes [7]. Faster hitting time and the ability to traverse arbitrary structures make QRW a good basis for a variety of quantum algorithms, such as the one for finding triangles in a graph [8], calculating Boolean formulas [9], quantum unsupervised machine learning [10], quantum neural networks [11], quantum signature schemes [12], and the quantum random walk search algorithm (QRWS) [13]. The latter algorithm is used to search different topologies, such as simplex and star, graphs [14], trees [15], square grids [4], and hypercubes [16]. Due to the diversity of tasks where QRW is used, a variety of experimental implementations have been considered, such as optical quantum computers [17], optical lattice [18], circuit QED [19], and ion traps [20].

Most quantum algorithms are constructed using two state systems—qubits. However, using qudits instead of qubits has many advantages. The most obvious of them is exponential growth of the databases, preserving the number of information carriers [21]. Other advantages of using qudits are that they are more robust to external noise quantum gates [22], have more secure semi-quantum [23] and quantum [24,25] cryptographic protocols, and increase the effectiveness of a variety of quantum algorithms. Examples of such algorithms are Shor's factoring [26], Grover's search [27], and quantum counting [28].

Qudits are also used in some quantum random walk-based algorithms, such as Boolean formula evaluation [9] and quantum unsupervised machine learning [10].

There are two main methods for generating arbitrary d-dimensional quantum gates used to construct algorithms with qudits—by Given's rotations [29] and by Householder reflections (HR) [30]. Decomposing quantum gates to HRs is quadratically faster and can be used for qudits with an arbitrary dimension [31]. It can be implemented effectively on various physical system such as ion traps [32] and photonic quantum computer [33].

In the case of linear ion traps cooled to very low temperature and individual addressing of the ions and construction of qudit gates, this requires some additional conditions to be met. For example, all Rabi frequencies of the ions interacting with the laser fields have to have the same time dependence. An HR can also be applied to qudits, when they are created by multipod system [27]. All qudit states are metastable levels of the ion and transitions between them are made through the ancilla state. This technique requires precise control of the laser field's parameters [32] (such as pulse shape and detuning). Some techniques for reducing the sensitivity to such error types already exists. For example, composite pulses [34] can be very useful in many applications. However, in some cases, composite pulses alone will not be enough to reduce the error to the appropriate level for the algorithm to work properly.

Studying the effects of noise in quantum information is necessary, as they have undesirable but inevitable effects on the quantum system, which leads to decoherence. The physical interactions that underlie the occurrence of noise depend on the implementation of the quantum computer. There are different types of errors that may occur, such as errors in the ion's state's population or errors in the quantum state's phase. The effect of quantum noise on the system is often studied by assuming that the quantum gates are imperfect, or by adding additional noise gates [29,35]. The noise lowers the performance of quantum algorithms and quantum communication. For example, in [36], it was shown how noise strongly affects communication schemes based on quantum teleportation. As another example, in Grover's algorithm decoherence reduces the probability to find the solution by increasing the needed iterations. When the number of errors in the quantum register, occurring during the algorithm's implementation, passes a certain threshold—it stops working entirely [37].

The quantum walk, as in any other quantum system, interacts with the outside medium. This leads to noise that impacts its performance. That is why there is a lot of research on how the quantum walk is affected by noise. We will give a few examples. In one paper [38], using a one-dimensional quantum walk, an array of quantum dots was studied, where the authors paid particular attention to how noise and inaccuracies in their circuit gates affect the quantum walk. In Chandrashekar et al. [39], they presented a detailed theoretical study of the impact of different types of errors on the quantum walk on line when the coin has different symmetries. It is also discussed how different types of errors affect the QRW implementation in nuclear magnetic resonance quantum information processor and for atoms in an optical lattice. Due to the importance of knowing the type and magnitude of noise in a quantum system to assess its negative impact, a method that can be used to sense different types of noise in one- and two-dimensional quantum walks was studied [40].

A comparison of how noise affects Grover's search and discrete QRWS algorithm was made [41]. Their numerical calculations suggest that QRWS is more robust against white noise. How continuous time QRWS on different topologies is affected by the noise is studied in the work by Chiang and Hsieh[42]. The authors also investigated the relation of certain quantum parameters and algorithm's robustness. The continuous time quantum search algorithm on hypercube with noise was studied [43]. The discrete time quantum walk search algorithm with systematic phase error in the walk coin was investigated [44]. The authors use as a coin only Generalized HR, and show that the algorithm has very low robustness in this case. They investigated the algorithm's stability against phase errors in the HR.

Currently, machine learning (ML) is a diverse and modern topic that is used for a variety of tasks that include speech recognition [45], image and video generation [46,47], and medical diagnostics [48]. The quantum analog of the classical ML is an object of high scientific interest too [49,50].

In our previous work [51], we discussed discrete time QRWS on a hypercube algorithm with a walk coin constructed by one generalized HR and a phase multiplier. Numerical simulations for some particular cases (coins consisting of 1, 2, and 3 qubits) and particular configuration of the parameters showed high robustness of the algorithm to inaccuracies in those parameters.

In the current work, we investigate the stability of the modified QRWS algorithm (constructed as in [51]) in the case of qudit coins with a size from 2 to 12. Linear and nonlinear functional dependencies between walk coin parameters are studied, and we show that some of those relations lead to high robustness of QRWS on hypercube. Here, we make a detailed analysis of the quantum algorithm's robustness to simultaneous variations in both the generalized Householder phase and an additional parameter introduced by the nonlinear functions mentioned above. We show that there is a two-dimensional region in the combined space spanned by those parameters that gives very high robustness to the algorithm. Next, we point out that the modified algorithm becomes more stable than the one with coin built by just HR (as in [44]). Most of the results here are based on numerical methods, such as Monte Carlo simulations and ML, in addition to various numerical statistical techniques used to analyze the modified algorithm's robustness for larger set of coin register size.

This paper is organized as follows: In Section 2, a particular modification of the QRWS algorithm has been described. First, in Section 2.1, a brief description of QRWS algorithm and its quantum circuit is given. Next, in Section 2.2, we show the definition of the HR and its use in the construction of the walk coin. The simulation's results of the QRWS algorithm's walk coin with arbitrary dimension using HR and an additional phase are explained in Section 3. In Section 3.1, our Monte Carlo method for studying QRWS algorithm is explained and a few examples with a qudit coin are given. Next, Section 3.2 shows functional dependencies derived by MC simulation used to make the QRWS algorithm more robust to inaccuracies in the parameters of the walk coin. Section 4 shows our main results in this paper. In Section 4.1, we analyze how the region of stability changes with increasing the coin size. Robustness against inaccuracies in the relation between parameters is studied in more depth in Section 4.2. Numerical calculations of the optimal value of the parameters in the functional dependence between the phases for different walk coin size are shown in Section 4.3, with included ML prognosis for larger coins. The paper finishes with a conclusion in Section 5.

## 2. Quantum Random Walk Search with an Alternative Walk Coin

### 2.1. Quantum Random Walk Search Algorithm—Quantum Circuit

Quantum random walk search is a quantum algorithm designed for searching an unsorted database with arbitrary topology. Because of the faster traversing of structures by the quantum random walk, QRWS is quadratically faster than its classical counterpart. The algorithm is probabilistic with the probability to find a solution depending on the node register size [13], topology of the searched structure [4,16], modification of the algorithm [16], and the number and position of the solutions [52,53]. The quantum circuit of the QRWS algorithm in the case of qudits is shown in Figure 1.

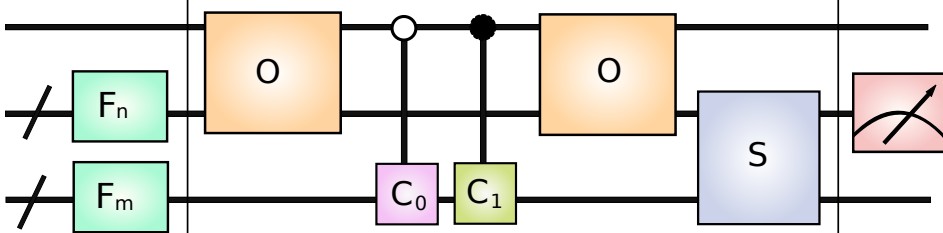

**Figure 1.** (Color online) Circuit of quantum random walk search algorithm.

The algorithm begins with preparing the initial states of node ($|x_{init}\rangle$) and edge ($|c_{init}\rangle$) registers in equal weight superposition and the control register qubit is in state $|0\rangle$. Such superposition can be achieved by applying the Discrete Fourier Transform operator on the coin and node registers when each of them is in state $|0\rangle$. In Figure 1, those operators are denoted by $F_m$ and $F_n$.

$$|c_{init}\rangle = \frac{1}{\sqrt{m}} \sum_{i=1}^{m} |i\rangle \tag{1}$$

$$|x_{init}\rangle = \frac{1}{\sqrt{n}} \sum_{i=1}^{n} |i\rangle \tag{2}$$

The dimensions of control, node, and coin registers are 2, $n$ and $m$, accordingly. The initial state of the whole algorithm has dimension $2mn$ and can be written as follows:

$$|\psi_0\rangle = |x_{init}\rangle \otimes |c_{init}\rangle \otimes |0\rangle \tag{3}$$

In the case of hypercube $n = 2^m$, so the dimension of the initial state is $m2^{m+1}$.

The algorithm continues by applying the algorithm's iteration a fixed number of times. The probability to find a solution is a periodic function of the number of iterations. Each iteration consists of the following steps:

First, an Oracle ($O$) is applied. Its goal is to recognize a solution and mark it, to ensure that the first coin $C_0$ (traversing coin) is applied to all non-solutions and the second coin $C_1$ (marking coin) is applied to all solutions. We have the following elements that are solutions $h_1, \ldots, h_\lambda$. The Oracle has dimension $2n$ (in the case of hypercube $2^{m+1}$) and the following matrix form:

$$O = \hat{I}_{2n} - \sum_{i=1}^{\lambda} [h_i, h_i] - \sum_{i=1}^{\lambda} [h_i - n, h_i - n] +$$
$$\sum_{i=1}^{\lambda} [h_i, h_i - n] + \sum_{i=1}^{\lambda} [h_i - n, h_i] \tag{4}$$

Here, $\hat{I}_{2n}$ is the identity operator with dimension $2n$, and $[r, c]$ denotes matrix element with value "1" positioned on the r-th row and c-th column. In this way, all register elements that are not solutions continue to stay in the second half of the register and all that are solutions are separated by moving them in the first half.

Traversing $C_0$ and marking coins $C_1$ are applied using controlled gates. Both controlled gates have block diagonal matrix representation and can be written as:

$$C_0 = \begin{pmatrix} \hat{I}_{m*n} & \hat{0}_{m*n} \\ \hat{0}_{m*n} & C_0 \otimes \hat{I}_n \end{pmatrix} \tag{5}$$

$$\mathcal{C}_1 = \begin{pmatrix} C_1 \otimes \hat{I}_n & \hat{0}_{m*n} \\ \hat{0}_{m*n} & \hat{I}_{m*n} \end{pmatrix} \tag{6}$$

where $\hat{0}_{m*n}$ is a matrix with dimension $m*n$ with all elements equal to zero. Hence, the simultaneous action of both controlled coins can be written as sequential applying of both operators:

$$\mathcal{C}' = \mathcal{C}_0 \cdot \mathcal{C}_1 \tag{7}$$

The operator $\mathcal{C}'$ acts on the node, coin, and control registers. This ensures applying the right coin on the nodes.

However, after acting with the coin operators, the whole circuit's register has to be returned to its initial form to ensure that the shift operator will make the desired walk. This should be performed by applying the inverse matrix of the Oracle. However, the Oracle is the inverse matrix of itself. This is why the Oracle has to be used a second time during each iteration of the QRWS algorithm. A description of the quantum circuit can also be seen in the following dissertation [54].

The shift operator $S$ defines the topology of searched structure by determining which nodes are connected by an edge. Depending on the probabilities of going in each direction (coming from the coin register state) the shift operator acts on the node register and executes the quantum walk. The shift operator can be written in the following way:

$$S = \sum_{i=0}^{2^m-1} \sum_{j=0}^{2^n-1} |i,j\rangle \langle i, g(i,j)|. \tag{8}$$

Here $m$ is a function of the elements of the node register $j$ and of the coin register $i$. This operator should be used at the end of each iteration. In case of a hypercube, the function $g(i,j)$ flips the $i$-th bit of vector $j$.

The algorithm's iteration (shown in Figure 1 of the quantum circuit with black rectangle) can be written as:

$$W = (\hat{I}_2 \otimes S) \cdot (O \otimes \hat{I}_m) \cdot \mathcal{C}' \cdot (O \otimes \hat{I}_m) \tag{9}$$

The state of the whole register after each iteration can be written as:

$$|\psi_{k+1}\rangle = W|\psi_k\rangle \tag{10}$$

The algorithm ends with a measurement of the node register. If the result is not one of the searched for elements, the algorithm is repeated.

The original QRWS for a hypercube [13] uses a Grover coin as traversing coin and have probability to find solution approximately $1/2 - \mathcal{O}(1/2^m)$, where $m$ is the size of the edge register (and, respectively, node register has $n = 2^m$ states). The number of iterations $k$ of the QRWS algorithm for hypercube in case one solution is obtained analytically when the Grover coin is used [16]:

$$k = \left\lceil \frac{\pi}{2}\sqrt{2^{m-1}} \right\rceil \tag{11}$$

This formula can be used for a coin consisting of qubits or one qudit with arbitrary dimension.

The maximum probability to find a solution depends on the walk coin operator. In the next section, we will show the alternative construction of the walk coin using HR and a phase multiplier. A similar modification of the algorithm in the case of qubits is shown in [51].

*2.2. Walk Coin by a Householder Reflection and an Additional Phase Multiplier*

The traverse coin, on an undirected graph, can be constructed using one generalized HR with phase $\phi$ and one phase multiplier $\zeta$ in front of the coin, as is explained in [51]. The modified coin operator is:

$$C_0(\phi, \chi, \zeta) = e^{i\zeta}(I - (1 - e^{i\phi})|\chi\rangle\langle\chi|). \tag{12}$$

If the state used to build HR $|\chi\rangle$ is equal weight superposition, then the probability to go at each adjacent node is equal.

Here, we will study the dependence of the probability to find a solution for the phases $\zeta$ and $\phi$. We will show that properly chosen relation between them will lead to more robust QRWS regardless of the node register's number of states. This will increase the probability in all experimental implementations where HR can be performed efficiently.

Two simple examples of the coin operator $C_0$ with different values of angles are the Grover coin when $\phi = \zeta = \pi$ (high probability) and the identity operator when $\phi = \zeta = 0$ (there is no random walk). In those cases, the probability to find solution of the QRWS depends only on the number of node register's states (i.e., $2^m$). However, in the general case the dependence is more complex:

$$p(\zeta, \phi, m) = \frac{1}{2} - \mathcal{O}\left(\frac{1}{2^m}\right)f(\phi, \zeta). \tag{13}$$

To search for the hypercube with dimension $m$, the walk coin, in the general case, has to be made by qudits. If it consists of qubits—$m$ will be a power of two, for qutrits—power of three, and so on. It is easy to see from Equation (13) that the algorithm can be implemented in all cases when the size of the coin register is the same even if the qudit internal structure is different—for example 2 qutrits or one qudit of size 9. However, the experimental realization will differ depending on the structure of the coin register. Further in this work, we will consider the case when the coin register is built by only one qudit with arbitrary dimension. The algorithm's node register will consist of qubits.

## 3. Qrws with Qudit Coin Constructed by Householder Reflection

Here, we study QRWS on a hypercube with a qudit coin that has a dimension of up to 12 states. Using qudits instead of qubits gives us more detailed information how the algorithm's parameters change as the coin size increases. Even more, the fact that qudits are more robust against noise, will further increase the modified QRWS algorithm's stability.

*3.1. Monte Carlo Simulations of Qrws*

We carried out series of Monte Carlo simulations of QRWS algorithm with qudit coin. Our numerical code takes random values for both angles $\phi, \zeta \in [0, 2\pi]$, and simulates the quantum algorithm with walk coin constructed by a HR and a phase multiplier with those values. The coin size is prior known, so we will consider it as constant ($m = const$). After each run of the QRWS algorithm's simulation, the angles ($\phi, \zeta$) and the probability to find solution $p(\phi, \zeta)$ are stored.

Examples of Monte Carlo simulations for coins with dimensions 5 and 9 are shown in Figure 2a and Figure 2b, correspondingly. The darker color shows higher probability to find a solution, and lighter color—smaller $p(\phi, \zeta)$. In the case of coin size $m \geq 4$, there is a stripe with high probability to find solution. As $p(\phi, \zeta)$ increases, the width of the stripe decreases. In Appendix A in Figure A1, similar figures for coin size $m \in [2, 11]$ are shown. For $m \geq 4$, they have the same behavior as discussed above.

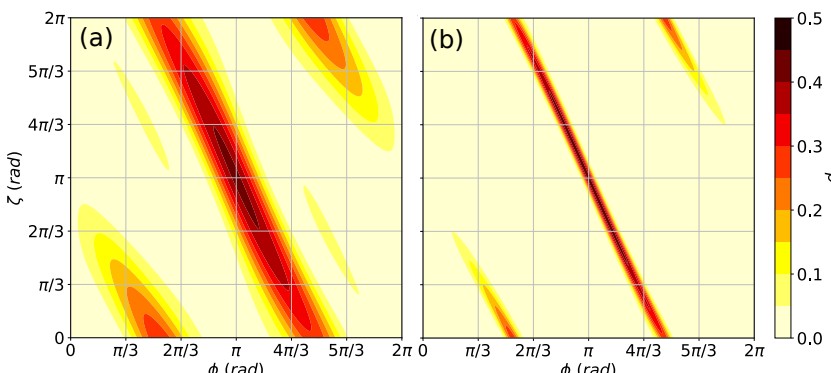

**Figure 2.** (Color online) Probability to find solution for coin sizes $m = 5$ (**a**) and $m = 9$ (**b**). The probability $p$ is plotted as a function of the angles $\phi$ and $\zeta$.

By using a ML model (described in Appendix F) trained on the Monte Carlo datapoints $(\phi, \zeta, p(\phi, \zeta))$ with coin size $m \in [2, 10]$, we make a prediction for $p(\phi, \zeta)$, $m \geq 11$. The results for coin size 11 and 16 are plotted in Figure 3b and Figure 3c, respectively. They show that ML also confirms the observation from the last paragraph for reducing the width of the stripe of high $p(\phi, \zeta)$ by increasing the coin size. In order to evaluate our ML model, in the case of QRWS with coin size $m = 11$, additional Monte Carlo simulations were performed—they were our validation set. The results are given in Figure 3a. If we compare them with the ML predictions Figure 3b, except for a slight blurring in the ML extrapolations explained in Appendix F, both figures show quite similar behavior.

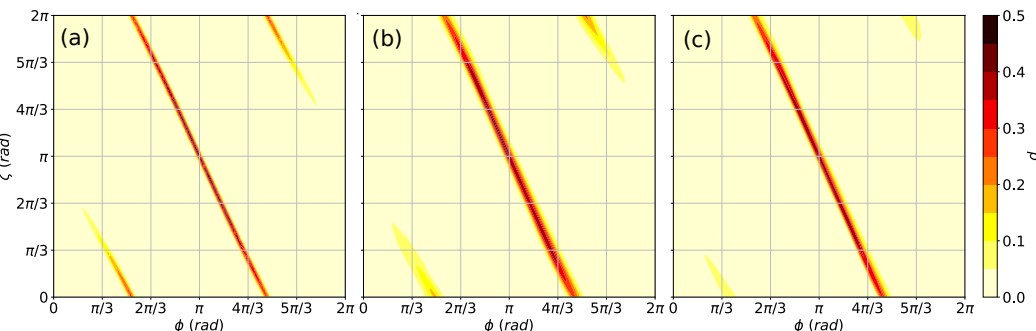

**Figure 3.** (Color online) Probability to find solution $p(\phi, \zeta, m)$ for coin sizes $m = 11$ (**a**,**b**) and $m = 16$ (**c**). The image (**a**) shows Monte Carlo simulation results and images (**b**,**c**) show the ML predictions.

The observation of stripe with high probability $p(\zeta, \phi, m)$, persisting for all studied coin dimension, point us toward defining a function $\zeta = \zeta(\phi)$, and the probability to find a solution becomes a curve $p(\phi)$ in the plane spanned by $\zeta$ and $\phi$:

$$p(\phi, \zeta, m) \longrightarrow p(\phi, \zeta(\phi), m = const) \equiv p(\phi) \tag{14}$$

The construction of such a dependence between coin parameters effectively eliminates one of the coin parameters, while allowing us to significantly increase the quantum algorithm's robustness, as will be shown in the next sections.

### 3.2. Robustness of the Coin for Different Functions $\zeta(\phi)$

Let $p_{max}$ be the maximum probability to find solution (for particular $m = const$) and $\phi_{max}$ is the value of $\phi$ where $p_{max}$ is achieved. Let the phase $\phi$ vary in the interval defined by $\Delta = (\phi_{max} - \varepsilon_-, \phi_{max} + \varepsilon_+)$, where $\varepsilon_-$ and $\varepsilon_+$ are different for different angle dependencies $\zeta(\phi)$ and coin sizes. In this case, the following equality holds:

$$p(\phi \in (\phi_{max} - \varepsilon_-, \phi_{max} + \varepsilon_+)) \cong p_{max} = p(\phi_{max}) \tag{15}$$

Our goal is to find function $\zeta(\phi)$ that gives the largest possible interval $\Delta$. This will make the QRWS algorithm more robust to variation of the parameter $\phi$.

Here, we will consider two types of functions. First, linear ones:

$$\zeta = -2\,\phi + 3\pi \tag{16}$$

and

$$\zeta = \pi \tag{17}$$

are considered because the former (Equation (16)) gives the best linear fit and the latter line (Equation (17)) shows the case when parameter $\zeta$ is constant and this is a representation of the case when during the experiment no special relation between $\phi$ and $\zeta$ is preserved. The case $\zeta = \pi$ was studied by Zhang et al. in [44].

The second type of function we have considered is:

$$\zeta = -2\,\phi + 3\pi + \alpha \sin(2\phi), \phi\epsilon[0, 2\pi]. \tag{18}$$

Such a non-linear function gives better results in comparison to the linear ones shown above. However, here we introduce another parameter $\alpha$. The best value of $\alpha$ depends on the coin size, but relatively good initial choice is $\alpha = -1/(2\pi)$, as will be shown later in this paper. Such a curve better approximates $p(\phi)$; however, sinus function would be more difficult to implement in the experiments.

An in depth study of the value of parameter $\alpha$ obtained by Monte Carlo and ML, namely $\alpha_{ML}$, is given in Section 4.3. Using ML for prediction of $\alpha$ is required due to the impossibility of simulating the QRWS algorithm for large coin sizes.

The probability to find a solution for the functions specified above, in the case of coin size 5 and 9, is shown in Figure 4. Numerical simulations of the probability $p(\phi)$, where different relations between coin parameters are used: (16) (red dot-dashed line), (17) (teal dashed line), and Equation (18) with $\alpha = -1/(2\pi)$ (blue dotted line) were made, and the value $\alpha = \alpha_{ML}(m)$ is predicted by ML (solid green line). The lines given by Equations (16)–(18) have axial symmetry around the vertical line $\phi = \pi$.

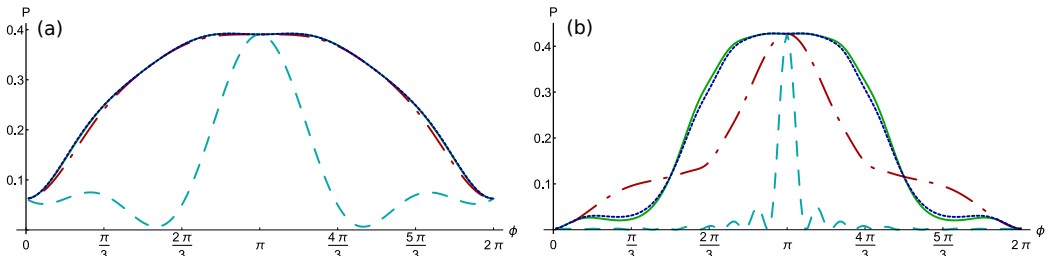

**Figure 4.** (Color online) The probability to find a solution for coin sizes 5 (**a**) and 9 (**b**) in the case of different functions $\zeta(\phi)$. The red dot-dashed line corresponds to Equation (16), the blue dotted line—to Equation (18) with $\alpha = -1/(2\pi)$, the teal dashed line—to Equation (17), the solid green line—to Equation (18) with $\alpha$ obtained by ML.

In Appendix B in Figure A2, the probabilities $p(\phi, m), m \in [2, 12]$, for different functions $\zeta(\phi)$ are shown (the same as in Figure 4).

A comparison between simulation results for our validation set, consisting of $p(\phi, m = 11)$ and $p(\phi, m = 12)$ of QRWS algorithm and predictions of the trained neural network for different relations $\zeta(\phi)$, is made to verify the reliability of the DNN extrapolations given in the paper. In Figure 5a lines are drawn, obtained from Monte Carlo simulations and in Figure 5b, which show the predictions of the ML model. It can be clearly seen that the curves on both figures show the same behavior as, for example, the similar shape of the different lines (the solid green, dotted blue, and dot-dashed red corresponding to $\alpha = \alpha_{ML}, \alpha = -(2\pi)^{-1}, \alpha = 0$ in Equation (18), respectively, and the dashed teal line—to

Equation (17)). However, there are some asymmetries in the curves from the ML model. The lines in Figure 5c represent predictions of the neural network for QRWS algorithm with walk coin size $m = 16$. By comparing both ML extrapolations for $m = 11$ (Figure 5b) and $m = 16$ (Figure 5c), it is visible that all lines became steeper and there is an increase in the high probability $p(\phi, m)$ plateau with the increase of $m$, an observation supported by the MC simulations for lower coin register size (Figure 4).

Due to higher computational time and memory demands, for coin size 12, we calculated fewer points for each of the probability curves (see Appendix B). However, those points were enough to confirm our predictions:

(1) Reducing the width of the curves with increasing of $m$;
(2) The suggested nonlinear dependence between angles (18) gives the highest stability of the algorithm. Worst performance is when $\zeta = const$.

Those results give us the premise to argue that the QRWS built with an alternative walk coin, proposed in [51] and thoroughly studied in this work, will give more robust implementation of the quantum algorithm for large coin register.

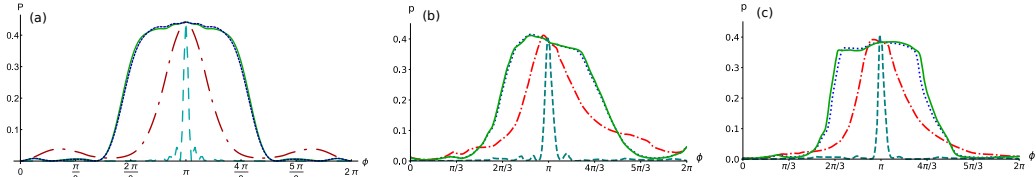

**Figure 5.** (Color online) The probability to find a solution $p(\phi, m)$ for different functions $\zeta(\phi)$: the red dot-dashed line corresponds to Equation (16), the blue dotted line—to Equation (18) with $\alpha = -1/(2\pi)$, the teal dashed line—to Equation (17), the green line—to Equation (18) with $\alpha$ obtained by ML. (**a**) shows results from Monte Carlo simulations for coin size $m = 11$, (**b**)—ML predictions for $m = 11$, and the one on (**c**)—ML predictions for $m = 16$.

Based on our numerical results, we predict that with increasing the coin size, the curves for probability $p(\phi)$ given by Equation (18) with $\alpha = -1/(2\pi)$ and $\alpha = \alpha_{ML}(m)$ will become indistinguishable. Those curves will become steeper and closer to trapezoids.

Simulations of the algorithm with coin constructed using relation (18) result in higher $p_{max}$ in comparison with the standard construction (with constant phase factor $\zeta = \pi$ as in Equation (17)). In Table 1, are shown the maxima of probability $p(\phi, \zeta(\phi), m)$, for each of the functions $\zeta(\phi)$ (Equation (16), Equation (17), and Equation (18) with $\alpha = -1/(2\pi)$ and $\alpha = \alpha_{ML}(m)$) and coin size $\phi\epsilon[0, 2\pi]$.

**Table 1.** Maxima of probability to find solution by QRWS for different coin size when relation (17) is used are shown in the first row. Analogically, $p$ corresponding to Equation (18), with $\alpha = 0$ is shown in the second row, and with $\alpha = -1/(2\pi)$—on the third row. The fourth row corresponds to Equation (18) with $\alpha$ obtained by ML.

| Line\Coin Size | 4 | 5 | 6 | 7 | 8 | 9 | 10 | 11 |
|---|---|---|---|---|---|---|---|---|
| Equation (17) | 0.3906 | 0.4137 | 0.4117 | 0.4022 | 0.4344 | 0.4272 | 0.4334 | 0.4414 |
| Equation (18) & $\alpha = 0$ | 0.3906 | 0.4137 | 0.4117 | 0.4022 | 0.4344 | 0.4272 | 0.4334 | 0.4414 |
| Equation (18) & $\alpha = (-2\pi)^{-1}$ | 0.3921 | 0.4137 | 0.4117 | 0.4082 | 0.4344 | 0.4279 | 0.4354 | 0.4414 |
| Equation (18) & $\alpha = \alpha_{ML}$ | 0.3921 | 0.4137 | 0.4117 | 0.4093 | 0.4344 | 0.4277 | 0.4344 | 0.4414 |

As the coin size increases, the probability to find the solution becomes higher. In the case of Equations (16) and (17) for coin size greater than 3 the maximum is always at the point with coordinates $\phi = \pi$ and $\zeta = \pi$ (corresponding to Grover coin). However, $p_{max}$ is not always at this point if Equation (18) is used. For example, when $\alpha = -1/2\pi$, relation (18) becomes $\zeta = -2\phi + 3\pi + (-1/(2\pi))\sin(2\phi)$. It gives higher $p(\phi)$ in comparison to the

Grover coin for coin sizes $m = 4, 7, 9, 10$. In the particular case of $m = 7$ the maxima are at points $\varphi_{max}^{(1)}$ and $\varphi_{max}^{(2)}$, where $\varphi_{max}^{(1)} = 2.7925$ and $\varphi_{max}^{(2)} = 3.4907$ (with numerical uncertainty $\Delta_\varphi = 0.0349$). From Table 1 it is clear that the increase is significant.

The results above show that the numerical methods used in this section can describe, important for us, properties of QRWS algorithm sufficiently well. That is why we use them to obtain the main results for the algorithm's robustness.

## 4. Numerical Results

### 4.1. Region of Stability for Different Coin Sizes

During the experiments, there always are unavoidable time variations of the experimental setup's parameters, such as in the frequency and the shape of the laser pulses controlling ions in the trap. In order to investigate the robustness of QRWS algorithm's implementation we will study the half-length $\varepsilon = (\varepsilon_- + \varepsilon_+)/2$ of the interval $\Delta$ (defined in Section 3.2). This quantity will allow us to quantitatively evaluate the stability of the algorithm and make comparisons for different register sizes and functional dependencies between the coin parameters.

Here we show results for the width $\varepsilon$ of each of the functions (16), (17), and Equation (18) with $\alpha = -1/(2\pi)$ and $\alpha = \alpha_{ML}(m)$, where $m \in [2, 16]$. The solid lines correspond to results from MC simulation, and dashed lines—to prognosis given by ML. In Appendix F, an explanation of ML methods used and their drawbacks are given. Different colors and markers correspond to different dependence between phases in the walk coin. The red curve with 4-pointed star marker corresponds to Equation (16), the teal with 3-pointed star marker—to Equation (17), and the blue with 5-pointed star marker and the green with 2-pointed star marker—to Equation (18) with $\alpha = -1/(2\pi)$ and $\alpha = \alpha_{ML}(m)$ correspondingly. We remark that MC points for $m = 11$ were not enough to obtain good ML estimation of $\alpha$. This is why we used $\alpha_{ML}(m = 11)$ predicted with neural network model in QRWS algorithm simulation. This approach gives significantly better estimation for $\varepsilon$ in the case $m = 11$. In Figure 6 are shown examples for the values of $\varepsilon$ in the cases when $p(\phi \in (\phi_{max} - \varepsilon_-, \phi_{max} + \varepsilon_+)) \geq 0.9 \times p_{max}$ and analogically for $0.7 \times p_{max}$:

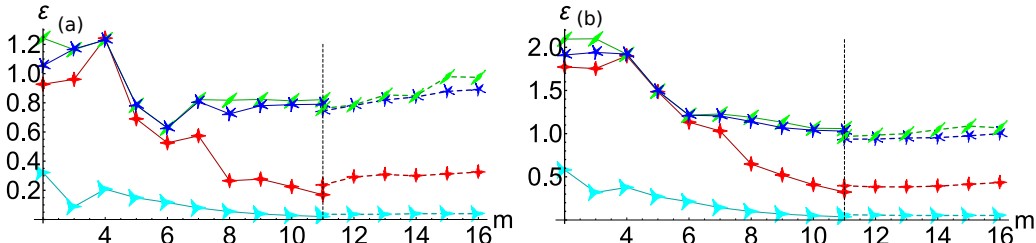

**Figure 6.** (Color online) Change of the area width $\varepsilon$, having probability to find solution equal to percentage of the maximum probability $p_{max}$, with increase of the coin size $m$. (**a**) shows the case for $90\% \times p_{max}$ and (**b**)—for $70\% \times p_{max}$. The red curve with 4-pointed star marker corresponds to Equation (16), the teal with 3-pointed star marker—to Equation (17), and the blue with 5-pointed star marker and the green with 2-pointed star marker—to Equation (18) with $\alpha = -1/(2\pi)$ and $\alpha = \alpha_{ML}(m)$ correspondingly. The solid lines correspond to values obtained by MC simulations, dashed to—results prognosed by ML.

The value of $\varepsilon$, depends on the coin size and on the function used to construct the coin. Monte Carlo simulation of QRWS using coins of size $m \leq 3$ have completely different behavior compared with the larger coins as can be seen in Appendix A in Figure A1. For those cases, the Grover coin gives very low probability to find solution. This is the reason to exclude MC simulations of QRWS with coin size $m \leq 3$ from further considerations.

In case of Equation (17) and for $m = 4$ the length $\varepsilon$ is low and it continues to decrease with increasing coin size. The curve from Equation (16) for small coins have high value of the width $\varepsilon$, however it decreases fast with increasing the coin size. For large coin we



predict that they will have similar behavior and Equation (17) will act like Equation (16) for smaller coin. So, the linear approximation is not good enough.

Equation (18) with $\alpha = 0$ coincides with Equation (16). However, other values of $\alpha$ are much more promising. The probability $p$ for coins constructed with functional dependence $\zeta(\phi)$, given by Equation (18), did not seem to decrease if the coin size increases. The curve corresponding to $\alpha = \alpha_{ML}(m)$ obtained by ML gives a slightly larger $\varepsilon$ and shows slightly better behavior with increase of the coin size compared to $\alpha = -1/(2\pi)$. The value of $\varepsilon$ deceases between 3 and 6, and remains approximately the same thereafter ($\varepsilon \simeq \pi/4$) as can be seen in Figure 6. The behavior of $\varepsilon$ is slightly different when we evaluate it for different percentage of the maximum probability. Two examples for $0.9 \times p_{max}$ and $0.7 \times p_{max}$ are shown in Figure 6.

Based on Monte Carlo simulations, we predict that $\varepsilon$, when $p(\phi \in (\phi_{max} - \varepsilon_{-}, \phi_{max} + \varepsilon_{+})) \simeq 0.9 \times p_{max}$, and for coin constructed according to Equations (17) and (16), will decrease with increasing the coin size and for large coin $\varepsilon \longrightarrow 0$. However, when using Equation (18) with $\alpha = \alpha_{ML}$, $\varepsilon$ will remain constant with increasing the coin size. The interval $\varepsilon$ with Equation (18) and $\alpha = (-1/(2\pi))$, will become closer to $\varepsilon$ corresponding to $\alpha = \alpha_{ML}$ and for large coins $\alpha_{ML} \longrightarrow (-1/(2\pi))$. For lower percentage of the maximum height (for example $0.7 \times p_{max}$), for all those functions $\varepsilon$ will decrease slowly with increasing the coin size and for large coin they will converge to a fixed value. The observations given above are supported by ML simulations, when we consider some drawbacks of our ML models, explained in Appendix F.

The analysis above gives us the base to assert that the half-width $\varepsilon$, did not decrease with increasing the coin register size if Equation (18) with optimal parameters is used. This indicates that such modification of QRWS algorithm retains its high robustness even for large coin size. Furthermore, that becomes very important when the registers reach several qudits enough for solving practical problems.

*4.2. Analysis of Algorithm'S Robustness*

The proposed in the work optimization of the QRWS algorithm is based on walk coin parameterized by two phases (see Equation (12)). We have showed in Section 3.2 that, if an optimal relation between them is maintained, the algorithm becomes more robust. Here, we will support this statement by numerical analysis of the stability of probability $p$ to uncertainties in the walk coin parameters $\phi$ and $\alpha = \alpha(\phi, \zeta)$. The latter expression is derived by solving Equation (18) for $\alpha$.

In Figure 7, simulations of the probability to find a solution $p(\phi, \alpha, m)$ are shown as a function of the angle $\phi$, the coin size $m = 5$ (in Figure 7a), $m = 9$ (in Figure 7b), and the parameter $\alpha$ of Equation (18).

For walk coin of size $m \geq 4$ there is a central plateau with high values of the probability $p(\phi, \alpha, m)$. The lightest color area corresponds to $p > 0.95 \times p_{max}(m)$, the second contour is at $p = 0.9 \times p_{max}(m)$, etc. The values of $p_{max}(m)$ are given in Table 1. The plateau is relatively wide not only in the horizontal axis (along $\phi$), but in $\alpha$ direction too. Furthermore, this behavior remains the same for all simulated QRWS schemes with walk coin size up to 10, as could be verified from the figures illustrating the function $p(\phi, \alpha, m), 2 \leq m \leq 10$ in Appendix D. The above observation shows that the gere proposed construction of the coin leads to a stable quantum algorithm against small deviations in both $\phi$, and the parameter $\alpha$ introduced by our scheme.

The horizontal lines on both images of Figure 7 mark specific values for the angle $\alpha$. The dashed gray, dash-dotted blue, and the solid green lines in Figure 7 correspond to $\alpha = 0$, $\alpha = -1/(2\pi)$, and $\alpha = \alpha_{ML}(m)$ (given in Table 2), respectively. The latter two lies close one to the other and both lie in regions with high probability p for all the simulated QRWS algorithms with walk coin size $m \geq 4$. This shows that the value $\alpha = -1/(2\pi)$ is relatively good and could be used in practice, even if the optimal value of $\alpha$ is not known precisely for quantum algorithms with even larger registers.

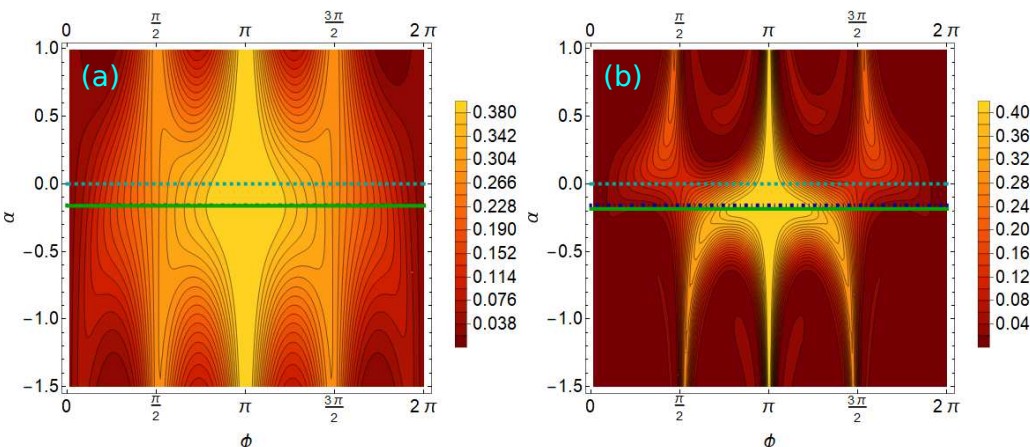

**Figure 7.** (Color online) Probability of QRWS algorithm to find solution $p(\phi, \alpha, m)$ as a function of $\alpha$ and $\phi$ for walk coin of size $m = 5$ (**a**) and $m = 9$ (**b**). The horizontal lines represent simulation of QRWS algorithm with $\alpha = 0$—dashed gray line, $\alpha = -1/(2\pi)$—dash-dotted blue line, and the solid green line correspond to $\alpha_{ML}(m)$ given in Table 2.

To give quantitative proof for the increased robustness of the QRWS algorithm, we make estimation of the root-mean-square (r.m.s) deviation of the probability $p(\phi, \alpha, m)$ for a fixed uncertainty in the parameters $\alpha$ and $\phi$, namely $\sigma_\alpha$ and $\sigma_\phi$. The numerical computations of the r.m.s. deviation of the probability $\sigma_p$ for each coin size $m$ are performed at fixed points—the nodes of a square grid in the plane $(\phi, \alpha)$. Each node is indexed by a pair of numbers $i$ and $j$. The values of the parameter $\phi$ are taken for 180 points equally spaced between 0 and $2\pi$, where $\phi^i = 2\pi/179(i-1), i = 1, \ldots, 180(\phi^0 = 0, \phi^{180} = 2\pi)$. Analogously, for $\alpha$ are taken 250 discrete values, so that $\alpha^j = \alpha^{min} + ((\alpha^{max} - \alpha^{min}))/249(j-1)$, where $j = 1, \ldots, 250$, $\alpha^{min} = -1.5$, and $\alpha^{max} = 1$. We do not assume any correlation between these parameters and calculate the quantity:

$$\sigma_p^{ij} = \frac{1}{p^{ij}} \sqrt{\left(\frac{\partial p^{ij}}{\partial \phi^i}\right)^2 \sigma_\phi^2 (\phi^i - \pi)^2 + \left(\frac{\partial p^{ij}}{\partial \alpha^j}\right)^2 \sigma_\alpha^2 (\alpha^j - \alpha_{ML}(m))^2} \qquad (19)$$

for each discrete point $\phi_i$ and $\alpha_j$. We center the data around the points $(\pi, \alpha_{ML}(m))$ which lie in the middle of the region giving the most stable implementation of the quantum algorithm. Here, $\alpha_{ML}(m), 2 \leq m \leq 10$ is the optimal value of the parameter $\alpha$ in Equation (18) obtained by fitting that expression to the numerical datapoints for QRWS with walk coin size m. Their values are given in Table 2.

The results for walk coin size 5 (Figure 8a) and 9 (Figure 8b) are graphically represented in a logarithmic scale. The actual values of the root-mean-square deviations $\sigma_\alpha$ and $\sigma_\phi$ depend on the particular physical realization of the algorithm, thus we have performed the computations with a fixed conservative value for both r.m.s. deviations $\sigma_\alpha = \sigma_\phi = 0.1$. As can be seen from the figures, there is a relatively large central area where the r.m.s. deviation of the probability $\sigma_p$ is less than 0.01, reaching levels $\sigma_p \leq 10^{-4}$ in the innermost dark region. This behavior is the same for the quantum algorithm using a walk coin with five and nine states. The corresponding images in Figure A6 in Appendix E for coin of size $2 \leq m \leq 10$ show similar characteristics. The above analysis confirms that the alternative walk coin, constructed by a HR and an additional phase, improves the QRWS algorithm, not only to obtain high probability $p(\phi, \alpha, m)$ for large interval of $\phi$, but also to have extremely high stability to changes in both parameters of the coin ($\varphi$ and $\alpha$).

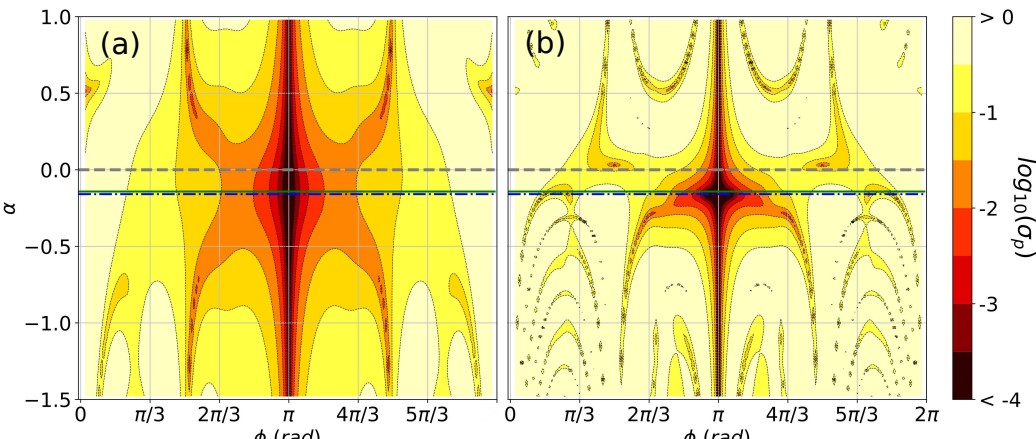

**Figure 8.** (Color online) R.m.s. deviation of the probability $p(\phi, \alpha, m)$, given by Equation (19), for walk coin of size $m = 5$ (on (**a**)) and $m = 9$ (on (**b**)). The dark central area represents high robustness of the quantum algorithm for small deviations of the algorithm's parameters. The dashed gray, dash-dotted blue, and the solid green lines correspond to $\alpha = 0$, $\alpha = -1/(2\pi)$, and $\alpha = \alpha_{ML}$.

It is useful to study how the robustness of the proposed construction of the QRWS algorithm compares to the stability of QRWS when the walk coin is built according to Equation (17)—corresponding to the case when the parameter $\zeta$ is not controlled as in [44] (if $\phi = \pi$ the Grover coin is obtained). We have calculated the r.m.s. of the probability for the latter case:

$$\sigma_{p'}^i = \frac{1}{p'^i}\left|\frac{\partial p'^i}{\partial \phi^i}\sigma_\phi(\phi^i - \pi)\right|. \tag{20}$$

Here $\sigma_\phi = 0.1$ and $p'(\phi, m)$ is the probability to find a solution corresponding to the curves $\zeta = \pi$ given by Equation (17) (depicted with the dashed teal lines on both images of Figure 4).

In Figure 9, the ratios $\sigma_p/\sigma_{p'}$ are plotted at the logarithmic scale (similarly to the case of $\sigma_p^{ij}$ in Equation (19)) and the division is performed for every value of $\alpha_j$ (i.e., $\sigma_p^{ij}/\sigma_{p'}^i |_{j=const}$), where the number of points on the coordinate axes are $1 \leq i \leq 100$ and $1 \leq j \leq 250$. Again, each node of the grid defined by the variables $\phi$ and $\alpha$ is indexed by a pair of numbers $i$ and $j$. Here, $\phi_{min} \leq \phi \leq \phi_{max}$ and $-1.5 \leq \alpha \leq 1$. The angles $\phi_{min}$ and $\phi_{max}$ are chosen in such a way that the intervals in $\phi$ correspond to the high probability central peaks of $p(\phi, \zeta = \pi, m)$ given by Equation (17) and plotted in Figure 4 (teal dashed lines). Outside this region, the probability $p(\phi, \zeta = \pi, m)$ of the standard QRWS algorithm is very low and comparison of the robustness is not relevant to our study. In Figure 9a,b, the results are shown for coin size $m = 5$ and $m = 9$, respectively. We will make remarks on two important points. First, the alternative QRWS shows stability $\geq \mathcal{O}(10^2)$ in comparison to the most commonly used walk coin. Second, by comparing the two images for coin size five and nine in Figure 9 (all the images for $2 \geq m \geq 10$ from Figure A7 in Appendix E show in general the same behavior), it can be seen that the relative stability further improves with increase of the coin size. For practical realizations of QRWS, the registers should have a large number of qubits(qudits). Thus, the expected improvement of the relative stability of the algorithm for a large walk coin register would lead to more robust experimental implementation of the QRWS algorithm.

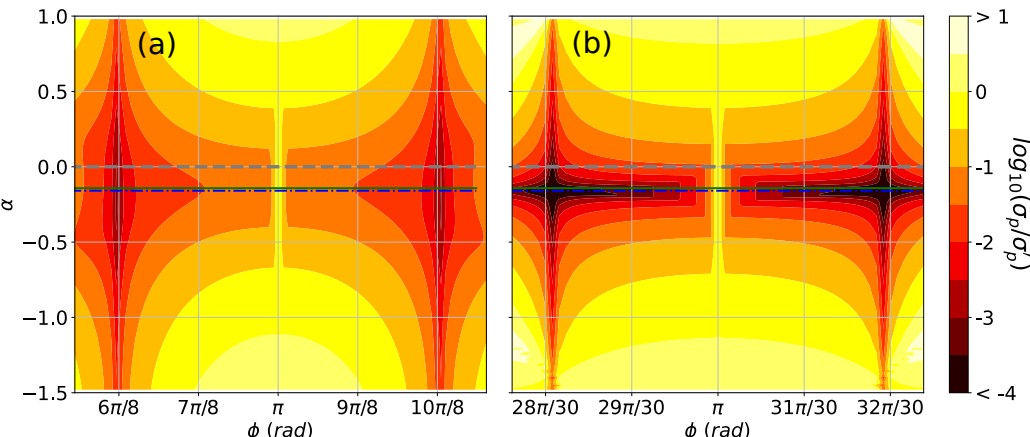

**Figure 9.** (Color online) Relative stability of the QRWS algorithm with walk coin size $m = 5$ (**a**) and $m = 9$ (**b**), given by the ratio of the root$-$mean$-$square uncertainty with and without the proposed optimization of the coin (Equation (19) and Equation (20), respectively). The chosen ranges of the parameter $\phi$ correspond to the high probability central peaks of $p(\phi, \zeta = \pi, m)$ (Equation (17)). The dashed gray, the dash-dotted blue, and the solid green lines are obtained from Equation (18) with $\alpha = 0$, $\alpha = -1/(2\pi)$, and $\alpha = \alpha_{ML}(m)$.

In this section, we studied in more detail the stability of the QRWS algorithm constructed with an alternative walk coin. The results show the existence of a large area around the optimal coin parameter values with very small r.m.s. deviation of the probability $p(\phi, \alpha, m)$ to uncertainties in both angles $\phi$ and $\alpha$. If we compare the stability of the proposed QRWS algorithm to the one with the commonly used walk coin, an increase in the relative robustness is observed for higher coin register size. The results show strong indication that, despite introducing an additional parameter $\alpha$ for the walk coin operator, and in this way adding more possibilities for noise, the algorithm is more stable than the one studied in [44], where the angle $\zeta = \pi$. These new results can be used for easier and robust implementation of QRWS algorithm.

### 4.3. Dependence between Alpha and Coin Size

In this section, we show results from in depth analysis of the parameter $\alpha$ introduced in our modification of QRWS algorithm for hypercube. Its value depends on the increase in the robustness of the studied modification of QRWS algorithm. Due to imperfections of the experimental setup, maximal theoretical probability to find a solution is unachievable. However, for many applications, probability equal to or higher than $0.9 \times p_{max}$ is sufficient. Based on this, we define the optimal value of alpha as the one that corresponds to a curve having the largest width at $p = 0.9 \times p_{max}$. The optimal values of $\alpha$ for each coin size $m$, namely $\alpha_{ML}(m), 2 \leq m \leq 20$ are given in Table 2. For coin size $2 \leq m \leq 11$, they are extracted from the numerical simulations' data and for $11 \leq m \leq 20$ (primed values)— from predictions of a deep network model, trained with the Monte Carlo datapoints for $2 \leq m \leq 10$. A detailed explanation of the DNN models used in the paper is given in Appendix F. In the case of coin of size 11, we give two relatively close values of $\alpha_{ML}$. They are derived independently, the first one from the MC simulations' data, and the second primed $\alpha_{ML}$ comes from ML predictions. The value of $\alpha_{ML}(11)$ obtained by MC has lower credibility because the simulated data points are much less than the ones for other values of $m$. The value $\alpha = -1/(2\pi) \simeq -0.159$ lies close to all $\alpha_{ML}(m), m \geq 4$ which justifies its use as a benchmark throughout the paper.

**Table 2.** Values of the parameter $\alpha_{ML}(m)$ in Equation (18) for QRWS algorithm's walk coin size $2 \leq m \leq 20$. The primed values are derived from ML predictions, and the remaining come from Monte Carlo simulations.

| *m* | 2 | 3 | 4 | 5 | 6 | 7 | 8 |
|---|---|---|---|---|---|---|---|
| $-\alpha_{ML}$ | 0.558 | 0.552 | 0.142 | 0.155 | 0.163 | 0.209 | 0.206 |
| *m* | 9 | 10 | 11 | 11 | 12 | 13 | 14 |
| $-\alpha_{ML}$ | 0.185 | 0.168 | 0.150 | 0.170$'$ | 0.179$'$ | 0.180$'$ | 0.203$'$ |
| *m* | 15 | 16 | 17 | 18 | 19 | 20 | |
| $-\alpha_{ML}$ | 0.225$'$ | 0.197$'$ | 0.205$'$ | 0.206$'$ | 0.216$'$ | 0.223$'$ | |

The values of $\alpha_{ML}$ extracted only from Monte Carlo simulations' data for QRWS with coin size $m = 2 \div 10$ are not enough to make any reasonable analysis of the behavior of this parameter for bigger coin size. However, by combining those results with the values of alpha obtained from the trained DNN models, we are able to make some predictions. To narrow the possible fitting functions, the following assumptions will be made. First, due to the completely different behavior of $p(\phi, m)$ for $m = 2, 3$ (see for example Figures A1 and A2), $\alpha_{ML}(m = 2)$ and $\alpha_{ML}(m = 3)$ are not included in the fitting procedure. Second, as explained in the text, the nonlinear term in Equation (18) could be considered to be a small perturbation to the linear functional dependence, so we expect that $\alpha$ will not become too large in absolute value for very large coin size m ($|\alpha(\infty)| < \alpha_\infty$, where $\alpha_\infty$ is a finite positive real number). An implication of the above statement is that if $\alpha_{ML}(m) \neq const$ (the simulations support this inequality), then $\alpha(m)$ should not be a linear function. Finally, the obtained values of $\alpha$ for different $m$, shown in Figure 10 (the teal dots are from MC data and the orange triangles represent ML predictions) are less than 20, so the fitting function has to be as simple as possible. We used a quadratic function and considered three scenarios. In the first scenario, we obtained $\alpha_1(m)$ (the dotted blue line in Figure 10) by suggesting that all points have equal weight. $\alpha_2(m)$ (the dot-dashed green line in Figure 10) is the fitting function, when the alpha values from the DNN model are considered to be less reliable (the fitting weights were set two times smaller) than the ones from Monte Carlo simulations data. The third, more well-grounded scenario, is an intermediate between the previous two. To obtain the function $\alpha_3(m)$ (the dashed red line in Figure 10) we have suggested that the ML predictions are more reliable for smaller coin size. This is due to the fact that the training of ML models is performed with data points with coin size $4 \leq m \leq 10$. To numerically achieve this, we have set weight coefficients for $\alpha_{ML}(m), m = 11 \div 20$ that decrease with increasing of $m$. The parameter values for the fitting functions (Equation (21)) in all of the studied scenarios are given in Table 3.

$$\alpha_i(m) = am^2 + bm + c, \qquad i = 1, 2, 3 \tag{21}$$

**Table 3.** Values of the parameters a, b, and c for the fitting functions $\alpha_i(m), i = 1, 2, 3$ (Equation (21)). The bolded last line represents the most reliable fit.

| | a | b | c |
|---|---|---|---|
| $\alpha_1(m)$ | $4.85 \times 10^{-5}$ | $-4.78 \times 10^{-3}$ | $-1.41 \times 10^{-1}$ |
| $\alpha_2(m)$ | $2.71 \times 10^{-4}$ | $-9.82 \times 10^{-3}$ | $-1.20 \times 10^{-1}$ |
| $\boldsymbol{\alpha_3(m)}$ | $\mathbf{1.17 \times 10^{-4}}$ | $\mathbf{-6.24 \times 10^{-3}}$ | $\mathbf{-1.35 \times 10^{-1}}$ |

All three lines explained above and shown in Figure 10, representing the behavior of $\alpha_{ML}$ as a function of the coin size in QRWS algorithm, meet the restrictions explained earlier in the section. The horizontal solid line plotted on the figure corresponds to $\alpha(m) = -1/(2\pi) = const$. All the functions $\alpha = \alpha(m)$ suggest a relatively large value of $\alpha_{ML}$ for coin registers with dimension even higher than 20. This translates to quantum

algorithm able to search in a database with an arbitrary topology consisting of $2^m$ entries. The functions $\alpha_i(m), i = 1, 2, 3$ given here, would allow physical implementation of the QRWS algorithm with increased robustness for any practically useful size of the database.

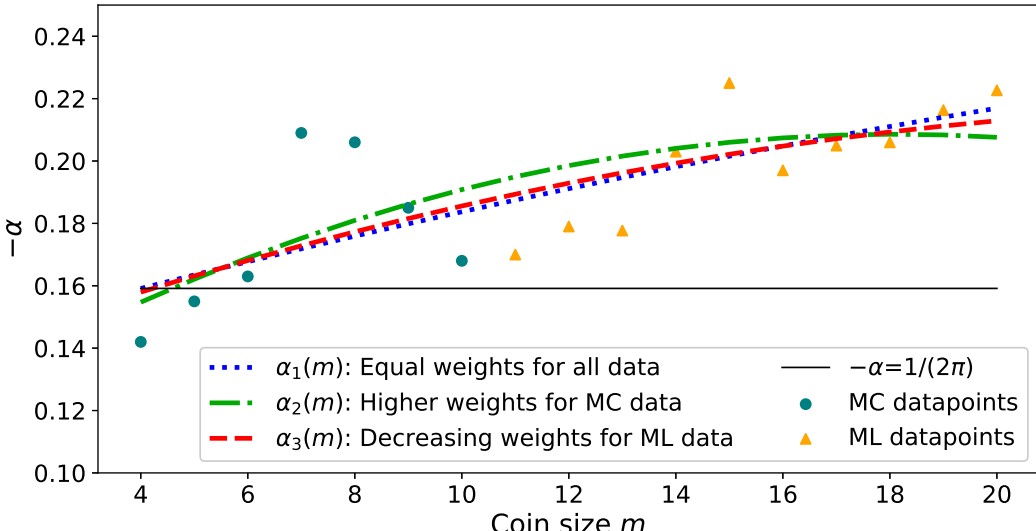

**Figure 10.** (Color online) Dependence of the parameter $\alpha_{ML}$ on the QRWS algorithm's coin size $m$. By fitting the values of $\alpha_{ML}$ obtained from MC simulations' data (teal dots) and ML predictions (orange triangles) three fits were derived. For the dotted blue line $\alpha_1(m)$, an equal weight of all data points was suggested. The dot-dashed green line $\alpha_2(m)$ is the fitting function, when the alpha values from the DNN model are considered to be ss reliable (the fitting weights were set two times smaller) than the ones from the Monte Carlo simulations' data. To obtain the function $\alpha_3(m)$ (the dashed red line) we have suggested that the ML predictions are more reliable for points close to the training area (i.e., the weight of $\alpha_{ML}(11)$ is higher than the weight of $\alpha_{ML}(20)$). The horizontal solid line figure corresponds to $\alpha(m) = -1/(2\pi) = const$.

In this section, the relation between $\alpha$ and the coin size of QRWS algorithm has been thoroughly studied. This parameter introduced as a factor giving the nonlinearity in the functional dependence between walk coin angles $\phi$ and $\zeta$, leads to an improvement in the stability of our modification of the random walk search algorithm. Here, by imposing a few reasonable restrictions to the possible fitting functions, three relations $\alpha = \alpha(m)$ were numerically derived.

**5. Conclusions**

In this paper, we studied a modified QRWS algorithm with improved stability. It uses an alternative walk coin constructed by a HR and an additional phase multiplier. In this work, the more general qudit walk coin is studied. An implementation of the modified QRWS algorithm with qudits has the potential to have numerous advantages. For example, qudits' robustness against noise, combined with the results presented in this paper, could make the quantum algorithm extremely stable to deviations in various experimental settings. Monte Carlo simulations of the QRWS algorithm for different sizes of the coin register were conducted. This allowed us to study in more detail the behavior of the proposed modification with increasing edge register's dimensions. It also enabled us to make extrapolations for the QRWS algorithm's stability at larger coin sizes. We have shown that, if a proper relation between the coin parameters is maintained, the quantum algorithm becomes much more robust, and this property remains true for every size of the quantum algorithm registers studied in the paper. We investigated the behavior of practically important quantities, such as the width of the area, resulting in high probability to find a solution and the maximum of this probability, for different coin parameters and relations between them. The dependence of nonlinear parameters (introduced in

the relation between coin phases) on the coin size was also studied by Monte Carlo and ML methods. Optimal functional dependencies and parameter values are derived for a walk coin of size up to one qudit with 11 states. Even more, by using ML methods, we made extrapolations of the mentioned quantities for larger sizes of the coin. From an analysis of the Monte Carlo data and ML predictions, we show that the QRWS algorithm, with the proposed alternative walk coin, demonstrates high robustness to deviations of its parameters. We calculated the stability of the proposed alternative construction of the QRWS algorithm. The numerical results show that there exists a wide area in the space spanned by the walk coin's parameters with extremely low root-mean-square uncertainty of the probability of QRWS algorithm to find a solution. The relative uncertainty of our modification of the quantum algorithm to the uncertainty of QRWS algorithm with a standard walk coin has been studied. We show that the former is more robust to coin parameter's deviations and also that the relative stability increases for a larger coin register.

**Author Contributions:** These authors contributed equally to this work. All authors have read and agreed to the published version of the manuscript.

**Funding:** This research was funded by Bulgarian National Science Fund under Grant *KP*-06-*M*48/2/26.11.2020.

**Data Availability Statement:** The datasets generated during and analyzed in the current study are available from the corresponding author on reasonable request.

**Acknowledgments:** The work on this paper was supported by the Bulgarian National Science Fund under Grant *KP*-06-*M*48/2/26.11.2020.

**Conflicts of Interest:** The authors declare no conflict of interest.

## Abbreviations

The following abbreviations are used in this manuscript:

| | |
|---|---|
| QRW | Quantum Random Walk |
| QRWS | Quantum Random Walk Search |
| HR | Householder Reflection |
| ML | Machine Learning |
| DNN | Deep Neural Network |
| Eq. | Equation |

## Appendix A. Monte Carlo Simulations for Different Coin Size

In Figure A1, Monte Carlo simulations' results are shown of the probability to find solution $p(\phi, \zeta, m)$ of the QRWS algorithm with a coin constructed by an HR and an additional phase multiplier. Simulation parameters are both phases $\phi$ and $\zeta$. Different images correspond to different size of the coin $m$. The images in the first row represent QRWS with coin size $m = 2, 3, 4$ (Figure A1a–c), on the second—with $m = 5, 6, 7$ (Figure A1d–f) and on the third—with $m = 8, 9, 10$ (Figure A1g–i).

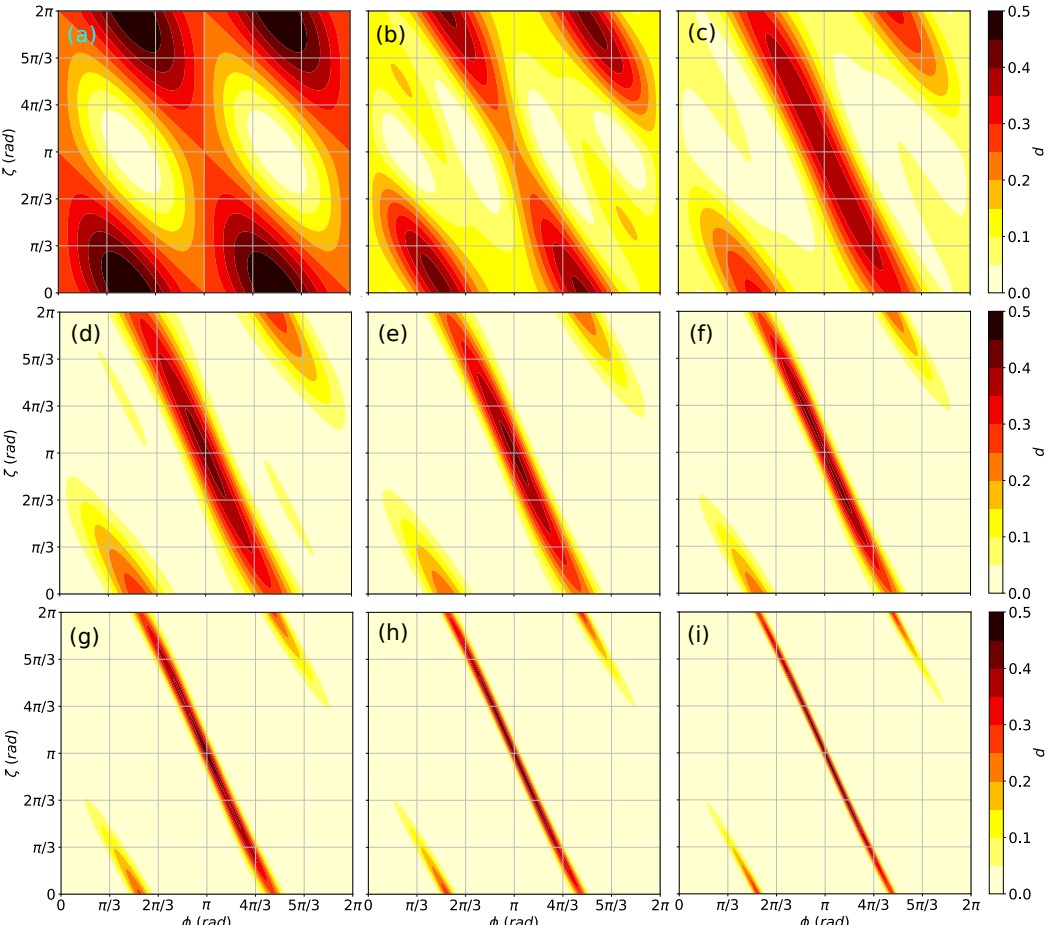

**Figure A1.** (Color online) Monte Carlo simulation of the probability to find solution *p* of QRWS algorithm with a coin constructed by a generalized HR and a phase multiplier. The coordinate axes correspond to the phases $\phi$ and $\zeta$. In the first row are images for coin size $m = 2, 3, 4$ (**a**–**c**), on the second—for $m = 5, 6, 7$ (**d**–**f**), and on the third—for $m = 8, 9, 10$ (**g**–**i**). Higher probability is shown with darker color.

For coin size 2, areas with high probability $p(\phi, \zeta, m)$ are chess-like arranged rhomboids. For coin size 3, those rhomboids begin to merge in the direction of the line $\zeta = -2\phi + 3\pi$. At coin size 4, the merged rhomboids became parallel stripes. From this point on, the stripes' width begins to decrease with increasing the size of the coin.

## Appendix B. Curves for Different Qudit Size

In Figure A2, the probability to find a solution $p(\phi, \zeta, m)$ for different relations between $\zeta$ and $\phi$ is shown. The red dot-dashed line corresponds to Equation (16) and the teal dashed line—to Equation (17). The blue dotted and the solid green lines show Equation (18) with $\alpha = -1/(2\pi)$ and $\alpha = \alpha_{ML}(m)$ correspondingly. In the first row are images for coin sizes 2, 3 (Figure A2a,b), on the second—4, 5 (Figure A2c,d), on the third—6, 7 (Figure A2e,f), on the fourth—8, 9 (Figure A2g,h), on the fifth—10, 12 (Figure A2i,j). The figure for coin size 12 is made with fewer points due to higher memory and CPU demands. For this case the simulated points are shown on the figure.

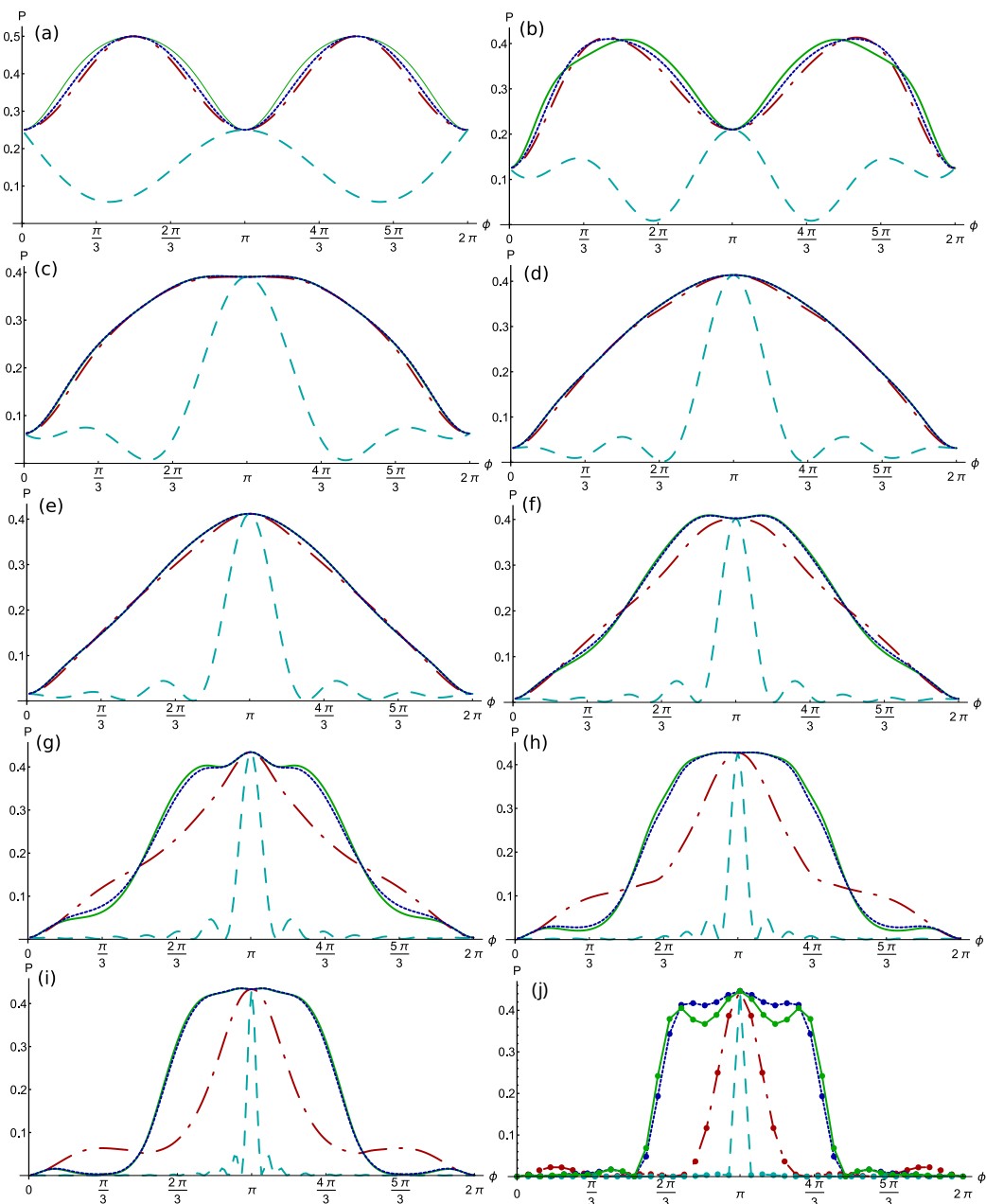

**Figure A2.** (Color online) Probability to find solution $p(\phi, m)$ for different coin size $2 \leq m \leq 10$ and $m = 12$. In the first row are images for coin size $m = 2, 3$ (**a,b**), on the second—for $m = 4, 5$ (**c,d**), on the third—for $m = 6, 7$ (**e,f**), on the fourth—for $m = 8, 9$ (**g,h**), and on the fifth—for $m = 10, 12$ (**i,j**). Different curves correspond to different relations between coin parameters: the red dot-dashed line—to Equation (16), the blue dotted line—to Equation (18) with $\alpha = -1/(2\pi)$, the teal dashed line—to Equation (17), the green line—to Equation (18) with $\alpha$ obtained by ML.

## Appendix C. Region of Stability for Different Coin Sizes—Practical Consideration

In Section 4.1, the interval where the, probability to find solution exceeds certain percentage of its maximum value has been studied. The more practical point of view is to investigate the interval $\varepsilon$ where the probability $p$ is greater than a fixed value $p > p_{LocM}(m)$. This can be used in experiments, when the goal is to reach a certain probability to find a solution after each run of the algorithm. Results of simulations for $p \geq 0.37$ and for $p \geq 0.31$ are shown on the left and the right sides of Figure A3. Different colors correspond to different dependences between the phases in the walk coin. The red curve with the 4-pointed star marker corresponds to Equation (16), the teal with the 3-pointed star marker—to Equation (17), and the

blue with the 5-pointed star marker and the green with the 2-pointed star marker correspond to Equation (18) with $\alpha = -1/(2\pi)$ and $\alpha = \alpha_{ML}(m)$ correspondingly.

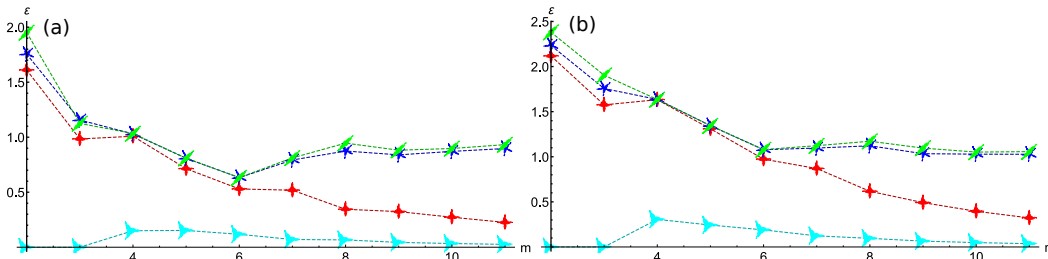

**Figure A3.** (Color online) Change of the width of the area with probability to find a solution greater than 0.37 (**a**) and greater than 0.31 (**b**). The red curve (4-pointed star marker) corresponds to Equation (16), the teal (3-pointed star marker) to Equation (17) and the blue (5-pointed star marker) and the green (2-pointed star marker) to Equation (18) with $\alpha = -1/(2\pi)$ and $\alpha = \alpha_{ML}(m)$ correspondingly.

When the walk coin is obtained by Equation (18) with $\alpha = \alpha_{ML}(m)$ or $\alpha = -1/(2\pi)$ we have better results. In Figure A3a $\varepsilon$ for all curves decreases until reaching size 6, then it begins to slowly increase until reaching coin size 11. However, more information is needed to say with certainty how they will behave with a larger coin register. In Figure A3b, $\varepsilon$ decreases fast until it reaches coin size 6 and then the decrease slows down.

The maximum probability to find solution increases with the coin size. Therefore, with the increase of the coin size, a fixed value of $p$ becomes a smaller percentage of $p_{max}$. That results in a different behavior of the curves compared to the ones in Figure 6. The lines in Figure A3 given by Equation (18) with $\alpha = \alpha_{ML}(m)$ and $\alpha = -1/(2\pi)$ did not converge to a fixed value and they continue to increase. From the above, we make the hypothesis that the robustness of the proposed algorithm modification increases if greater probability than fixed value is desired.

**Appendix D. Area of Stability of QRWS as Function of $\alpha$ and $\phi$**

In Figure A4, the probability $p(\phi, \alpha, m)$, when coin is constructed by a generalized HR and a phase multiplier (Equation (12)), and there is a relation between phases as in Equation (18), is shown. The walk coin depends on the phase $\phi$, the parameter $\alpha$, and the size of the coin $2 \leq m \leq 10$. Lighter colors correspond to higher $p$. The contours on the color plots show the probability $p$ in the range between 5% and 95% of $p_{max}$. The contour interval is 5%.

The more important case from practical point of view, analogically to Appendix C, is when we search for areas with probability $p$ greater than a fixed value. The images in Figure A5 show probability $p$ as function of $\phi$ and $\alpha$. However, contrary to Figure A4, here the contours are positioned at the same values of p for all coin sizes $m \in [2, 10]$. Contour values are between 0.037 and 0.37.

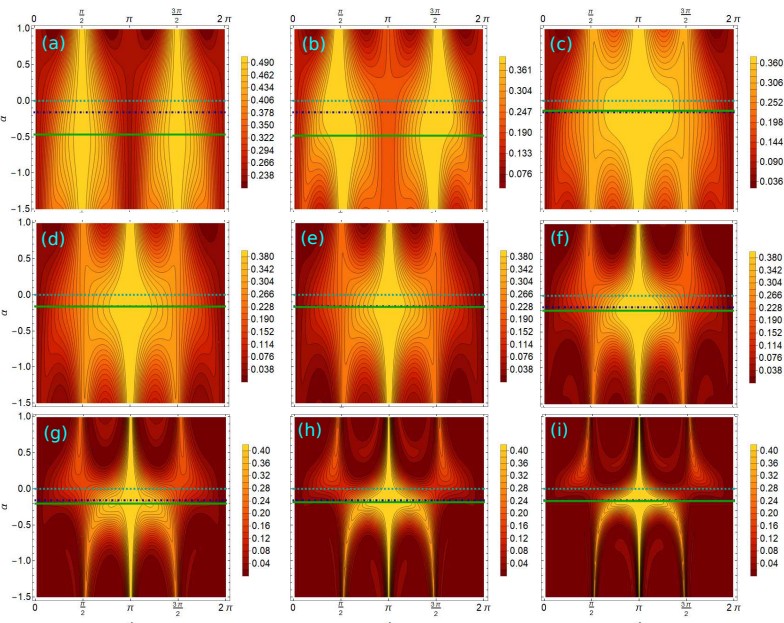

**Figure A4.** (Color online) Probability to find solution $p$ as function of $\phi$ and the parameter $\alpha$ in the non-linear Equation (18). The solid green line corresponds to $\alpha = \alpha_{ML}(m)$, the dash-doted blue—to $\alpha = -1/(2\pi)$ and dashed teal—to $\alpha = 0$. The lighter colors show higher probability and darker— lower. The scale is between 0 and $p_{max}$, and the contours are at intervals $5\% \times p_{max}$. The probabilities larger than $95\% \times p_{max}$ are depicted with the lightest color. Each picture corresponds to a different value of $m \in [2, 10]$. In the first row from left to right are given the images for $m = 2, 3, 4$ (**a**–**c**), on the second—$m = 5, 6, 7$ (**d**–**f**), and on the last—$m = 8, 9, 10$ (**g**–**i**).

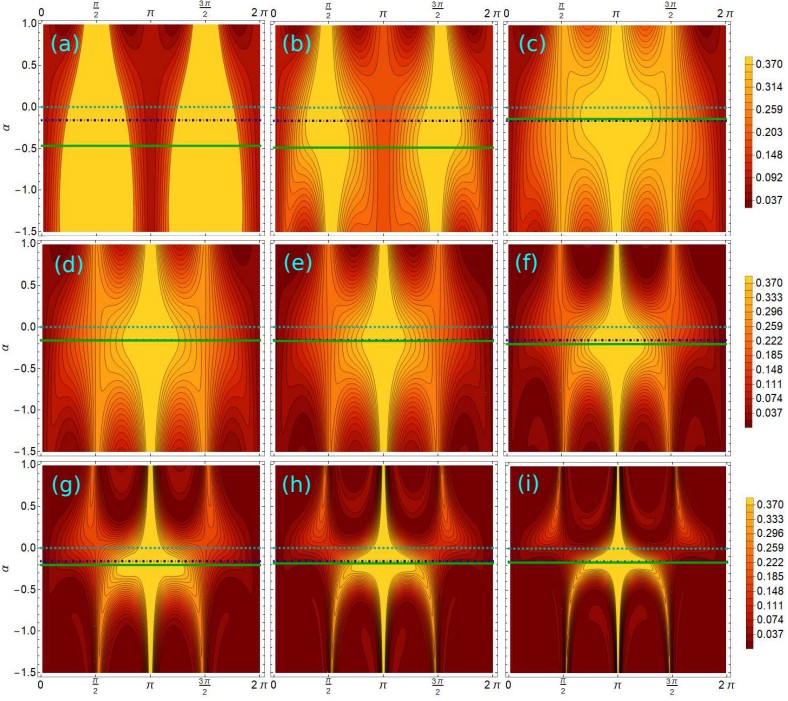

**Figure A5.** (Color online) Probability to find solution $p$ as function of $\phi$ and the parameter $\alpha$ in the non-linear Equation (18). The solid green line corresponds to $\alpha = \alpha_{ML}(m)$, the dash-doted blue—to $\alpha = -1/(2\pi)$ and dashed teal—to $\alpha = 0$. The lighter colors show higher probability and darker— lower. The scale is from 0 to $p_{max}$, and the contours are at fixed values of p (between 0.037 and 0.37). Each picture corresponds to a different value of $m \in [2, 10]$. In the first row from left to right are given the images for $m = 2, 3, 4$ (**a**–**c**), on the second—$m = 5, 6, 7$ (**d**–**f**), and on the last—$m = 8, 9, 10$ (**g**–**i**).

## Appendix E. Robustness of the Modified Quantum Random Walk Algorithm for Different Size of the Walk Coin

In Section 4.2, the stability of our proposed modification of the QRWS algorithm was investigated. A quantitative description of the root-mean-square deviation of the probability to find solution and the relative improvement in the stability have been graphically presented in the cases of walk coin size 5 and 9.

Here, in Figure A6, we give a complete set of results for the robustness of the QRWS algorithm, as defined in Equation (19), for coin of size $2 \leq m \leq 10$. It can be seen that, in all the images, there is a wide central area (darker colors) where the r.m.s. deviation of the probability $\sigma_p$ is extremely low for small changes in the parameters $\phi$ and $\alpha$.

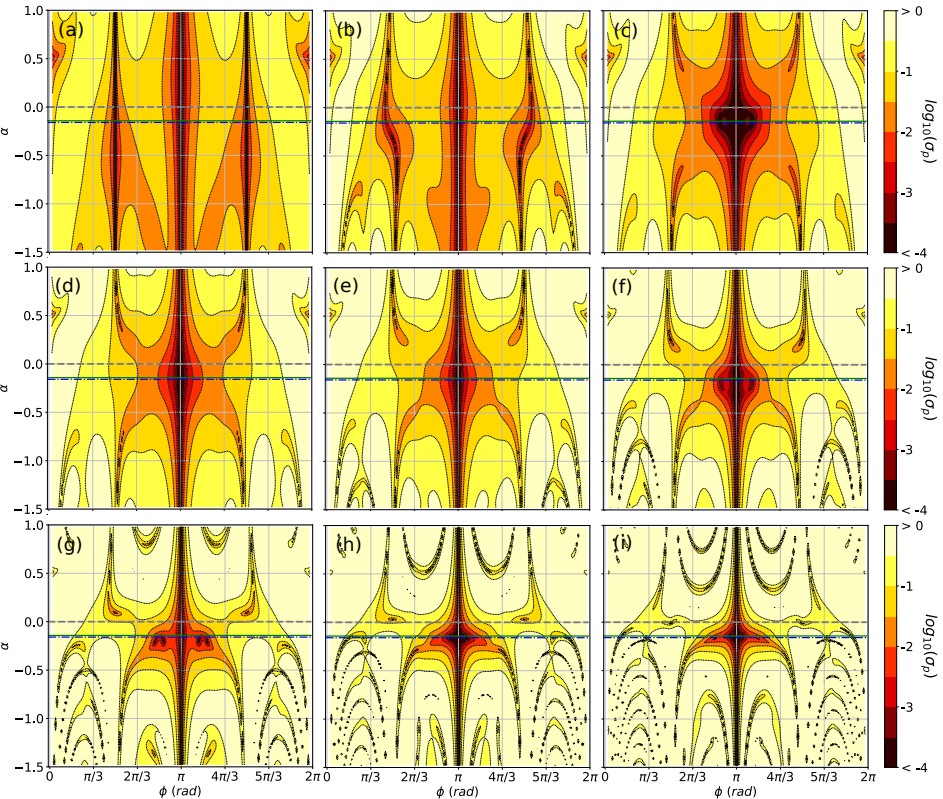

**Figure A6.** (Color online) Root-mean-square deviation of the probability $p(\phi, \alpha, m)$, given with Equation (19), for walk coin of size $m = 2, 3, 4$ (**a–c**), $m = 5, 6, 7$ (**d–f**), and $m = 8, 9, 10$ (**g–i**). The dark central area represents high stability region of the quantum algorithm for small deviations of the algorithm's parameters $\phi$ and $\alpha$. The dashed gray, the dash-dotted blue, and the solid green lines correspond to $\alpha = 0$, $\alpha = -1/(2\pi)$, and $\alpha = \alpha_{ML}$.

Analogously, in Figure A7 the numerical calculations' results for the ratio of the relative stability of the modified QRWS algorithm proposed by us are presented, to the one with standard construction of the walk coin. By increasing the coin size from $m = 2$ to $m = 10$, an expansion of the central dark area can be observed. This indicates that with increase of the coin size, the algorithm's relative stability continues to improve.

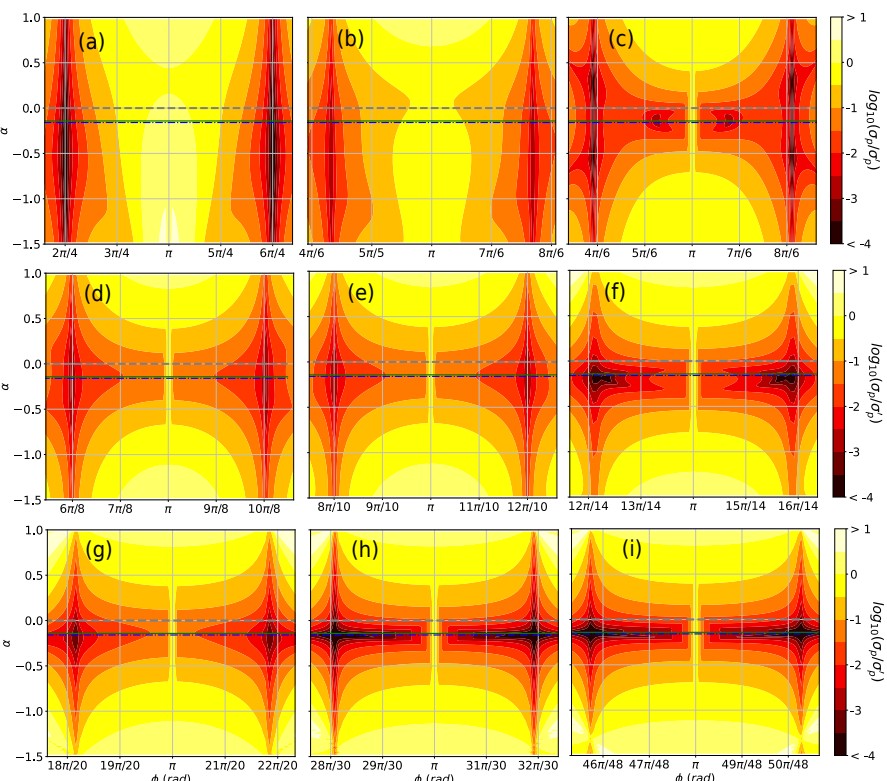

**Figure A7.** (Color online) Relative stability of the QRWS algorithm with walk coin size $m = 2, 3, 4$ (**a**–**c**), $m = 5, 6, 7$ (**d**–**f**), and $m = 8, 9, 10$ (**g**–**i**), given by the ratio of the root-mean-square uncertainty with and without the proposed optimization of the coin. The chosen ranges of the parameter $\phi$ correspond to the high probability central peaks of $p(\phi, \zeta = \pi, m)$ (Equation (17)). The dashed gray, dash-dotted blue, and the solid green lines correspond to $\alpha = 0$, $\alpha = -1/(2\pi)$, and $\alpha = \alpha_{ML}$.

**Appendix F. Deep Network Model and Machine Learning Predictions**

Simulations of quantum algorithms with a large number of qudits by classical computers are very demanding to computational resources. Simulating a QRWS algorithm for $m$-dimensional hypercube requires constructing a tensor product of the three registers used in the algorithm (control register of size 2, coin register of size $m$, and node register with dimension $2^m$), so the total state space is $m2^{(m+1)}$. The number of the algorithm's iterations needed to find solution also grows with increasing the coin size. Raising the size of both operators and the number of iterations leads to much larger computational times.

That requires an alternative approach to the task. In this paper, predictions of the studied quantum algorithm's parameters for bigger walk coin size, and, respectively, higher searchable space size, are made with ML methods.

Here, by Monte Carlo simulations of QRWS algorithm with different size of the coin register, we are able to obtain larger set of independent data that is used to train a ML model.

A feed-forward deep neural network (DNN) was built using the Monte Carlo generated datapoints for coin size $2 \leq m \leq 10$ to train it. Each datapoint consists of the walk coin parameters' values $\phi, \zeta, m$, and the corresponding MC results for the probability $p(\phi, \zeta, m)$ as labels. The neural network consists of L hidden layers, each with N neurons. The structure of the model used is shown in Figure A8. Next, the ML model is trained for a large number of epochs. Finally, it is used for predictions of the three-parameter function $p(\phi, \zeta, m)$, where $\phi \in [0, 2\pi], \zeta \in [0, 2\pi], m \geq 2$ (the results for coin size 11 and 16 are plotted in Figure 3). The values of parameters $\alpha$ and $\varepsilon$ defined in Section 4.1 are extracted by analyzing the above relations. Although our ML model gives relatively reliable predictions for several characteristics of the QRWS algorithm that are explained in the paper, others are not so good.

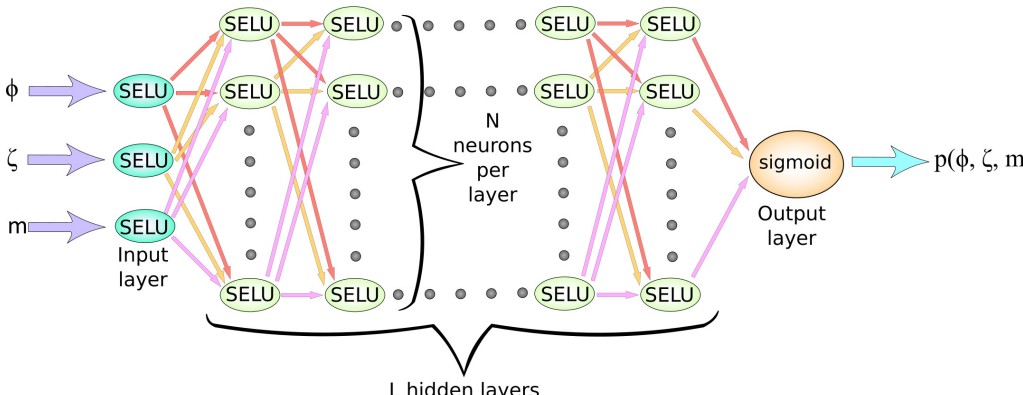

**Figure A8.** (Color online) Scheme of the Deep neural network used for prediction of the probability $p(\varphi, \zeta, m \geq 11)$.

It is a well known fact that using DNN for interpolation gives very good results. However, extrapolations away from the training region of the model result in inaccuracies in the predictions. In our case, they mainly manifest in small asymmetric widening of the central high probability strip of $p(\phi, \zeta, m)$ (Figure 3—middle and right images) that leads to some overestimation of the width $\varepsilon$, particularly for higher coin size m, seen in Figure 6. These deviations are more clearly visible when we focus on the highly nonlinear part of the function $p(\phi, \zeta, m)$ shown in Figure 3 for $m = 11, 16$.

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
