# Peer review of "A Machine Learning Study of High Robustness Quantum Walk Search Algorithm with Qudit Householder Coins"

_algorithms, doi:10.3390/a16030150_

Round 1

Reviewer 1 Report

The hard time limit, unfortunately, does not allow to understand all the details of the peer-reviewed article, but in general it can be noted that (i) the article looks very thorough; (ii) the issue of stability of the modified quantum random walk search algorithm was studied; time-consuming numerical calculations confirmed the superiority of the modified algorithm over the previously known one; the analysis performed allows choosing the optimal size of the qudit, which is important for practical implementation; (iii) the introduction and conclusion are written clearly, meaningfully and reflect the essence of the study.

I can recommend the article for publication practically in the form in which it is now, with a few remarks:

1. In the last paragraph of subsection 2.2, the authors state the following: `It is easy to see from Eq. (13), that the only thing that matters here is the total coin register size, not the way it is achieved – results of the simulations are the same for coin register consisting of 2 qutrits or one qudit with size 9.’ As far as I understand, the situation is somewhat more complicated. The algorithm used is designed in such a way that it does not have the ability to distinguish the internal structure of the qudit, and therefore the algorithm manipulates two qutrits in the same way as one 9-dimensional qudit. However, it does not follow from this that the simulation results will not differ in their subtle details. The authors did not investigate this issue, and this is not a fundamental flaw, but the authors simply did not have a theoretical tool for such a study. If I'm right, then maybe the authors will slightly refine the wording of their statement, making it less strict.

2. The definition of the parameter $d$ used in formula (4) is not found in the text.

Author Response

Dear Reviewer 1,

The answers to your questions and comments are in the attachment below.

Best Regards,

Hristo Tonchev and Petar Danev

Reviewer 2 Report

This manuscript present a quantum walk search algorithm with qudit Householder coins. The presentation is of high quality and the results looks promising. However I have several general question:

1. This manuscript entitles with "A machine learning study of high robustness quantum walk search algorithm with qudit Householder coins". Is "machine learning study" a suitable summary for the work presented in the manuscript? I feel most of the manuscript is about QRWS with qudit and only the last part is about some machine learning work?

2. If I understand correctly, for m dimension coin register the dimension of of node register is 2^m. That makes me feel the complexity is exponential for both space (to achieve 2^m one need exponential number of qubits/qudits in the node register) and time (controlled gate on 2^m is exponential). How about the qubit version QRWS? I think only better robustness cannot justify exponential complexity.

3. The DNN prediction is interesting but i wonder how to verify the prediction result is correct?  Are all available data are used to fit the model? Is there any available data used to validate/test the model in case of overfitting and prove the result from ML model is correct?

Author Response

Dear Reviewer 2,

The answers to your questions and comments are in the attachment below.

Best Regards,

Hristo Tonchev and Petar Danev

Reviewer 3 Report

The authors study the quantum random search algorithm with walk coin constructed by generalized Householder reflection and phase multiplier. Monte Carlo simulations in combination with supervised machine learning are used to find walk coins, that make the quantum algorithm more robust to deviations in the coins’ parameters. As a whole, the manuscript is smooth and accurate, and the work is fulfilling. Nevertheless, there exist many problems that should be addressed before further consideration.

1. At paragraph 2 of Section Introduction, the authors mention that “Qudits are also used in some quantum random walk based algorithms like Boolean formula evaluation [9] and quantum unsupervised machine learning [10]”. It is known that quantum walk based arbitrated quantum signature have also been proposed. It is better to cite the corresponding work [1] here for enriching the mentioned application of quantum walk with qudits.

[1] Arbitrated quantum signature scheme with quantum walk-based teleportation. Quantum Inf Process 18, 154 (2019).

2. Some expressions are not rigorous. For example, the expression “In [36], by a one-dimensional quantum walk, was studied an array of quantum dots” lacks of subject; In the expression “The authors pay particular attention to how noise and inaccuracies in their circuit gates affects the quantum walk”, the item ‘affects’ should be ‘affect’; In the description “In section 2 a particular modification of the quantum random walk search ….”, the expression “quantum random walk search” should be “QRWS” because it has been written with the abbreviation form; In the description “The initial state of the whole algorithm have dimension…”, the item ‘have’ should be ‘has’. It should be noted that there are many similar errors in the manuscript; All mathematical notations should be written in italics form; The expression “the curves for probability …will became indistinguishable” for became;

3. Please confirm the accuracy of Eq. (9) with consideration of the commutativity of matrix, the execution sequence of circuit and equation.

Some expressions are not uniform. For example, the use of “ML” and “machine learning”; the use of “equation” and “Eq”, and so on. Please note that the meaning of abbreviations.

4. In Figure 10, the legends can be added for clearly illustrating different curves.

5. The reference [2] below can be appropriately cited in the expression “The effect of quantum noise on the system is often studied by assuming that the quantum gates are imperfect or by adding additional noise gates [28,34]” in terms of its research on noise effect on quantum teleportation system.

[2] Parameterized Quantum Circuits for Learning Cooperative Quantum Teleportation[J]. Advanced Quantum Technologies, 2022, 5(9): 2200040.

6.It is also better for appropriately citing some recent quantum machine learning contributions as follows.

[3] Parameterized Hamiltonian Learning with Quantum Circuit, IEEE Transactions on Pattern Analysis and Machine Intelligence,doi:10.1109/TPAMI.2022.3203157 ,https://ieeexplore.ieee.org/document/9872139 (2022)

[4] Quantum Circuit Learning with Parameterized Boson Sampling, IEEE Transactions on Knowledge and Data Engineering, doi: 10.1109/TKDE.2021.3095103. https://ieeexplore.ieee.org/document/9477041,Volume: 35, Issue: 2, 01 February (2023)

Author Response

Dear Reviewer 3,

The answers to your questions and comments are in the attachment below.

Best Regards,

Hristo Tonchev and Petar Danev

Round 2

Reviewer 3 Report

The authors have addressed all my questions.